# CONSTRAINED DIFFUSION IMPLICIT MODELS

## ABSTRACT

This paper describes an efficient algorithm for solving noisy linear inverse problems using pretrained diffusion models. Extending the paradigm of denoising diffusion implicit models (DDIM), we propose constrained diffusion implicit models (CDIM) that modify the diffusion updates to enforce a constraint upon the final output. For noiseless inverse problems, CDIM exactly satisfies the constraints; in the noisy case, we generalize CDIM to satisfy an exact constraint on the residual distribution of the noise. Experiments across a variety of tasks and metrics show strong performance of CDIM, with analogous inference acceleration to unconstrained DDIM: 10 to 50 times faster than previous diffusion methods for inverse problems. We demonstrate the versatility of our approach on many problems including super-resolution, denoising, inpainting, deblurring, and 3D point cloud reprojection.

## 1 INTRODUCTION

Denoising diffusion probabilistic models (DDPMs) have recently emerged as powerful generative models capable of capturing complex data distributions (Ho et al., 2020). Their success has spurred interest in applying them to solve inverse problems, which are fundamental in fields such as computer vision, medical imaging, and signal processing (Tropp & Wright, 2010; Hansen, 2010). Inverse problems require recovering unknown signals from (possibly noisy) observations. Linear inverse problems, where the observations consist of linear measurements of a signal, encompass tasks like super-resolution, inpainting, and deblurring.

Existing methods that apply diffusion models to linear inverse problems face several limitations. First, many previous works require task specific training or fine-tuning (Li et al., 2022; Xie et al., 2023). Second, methods that use pretrained diffusion models often introduce many additional network evaluations during inference (Dou & Song, 2023; Zhu et al., 2024). Finally, popular diffusion inverse methods such as diffusion posterior sampling (Chung et al., 2022b) fail to exactly recover the input observations.

In this work, we propose constrained diffusion implicit models (CDIM), extending the inference acceleration of denoising diffusion implicit models (Song et al., 2021) to efficiently solve noisy linear inverse problems using a single pretrained diffusion model. Our method modifies the diffusion updates to enforce constraints on the final output, integrating measurement constraints directly into the diffusion process. In the noiseless case, this approach achieves exact recovery of the observations. For noisy observations, we generalize our method by optimizing the Kullback-Leibler (KL) divergence between the empirical residual distribution and a known noise distribution, effectively handling general noise models beyond the Gaussian assumption.

Our contributions are as follows:

- Accelerated inference: we accelerate inference, reducing the number of model evaluations and wall-clock time by an order of magnitude—10 to 50 times faster than previous posterior diffusion methods—while maintaining comparable quality.
- Exact recovery of noiseless observations: we can find solutions that exactly match the noiseless observation.
- General noise models: we extend the CDIM framework to accommodate arbitrary observational noise distributions through distributional divergence minimization, demonstrating effectiveness given non-Gaussian noise, such as Poisson noise.

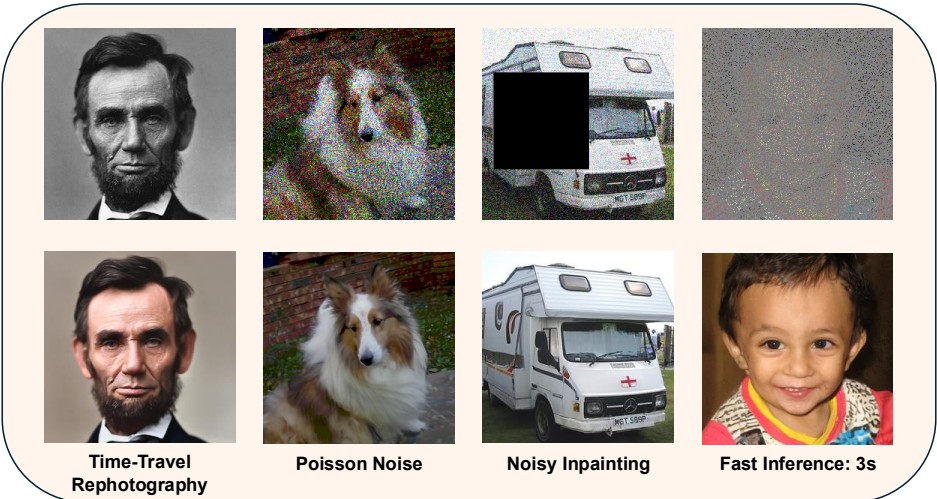

Figure 1: We show several applications of our method including image colorization, denoising, inpainting, and sparse recovery. We highlight the fact that we can handle general noise distributions, such as Poisson noise, and that our method runs in as little as 3 seconds.

## 2 RELATED WORK

Diffusion methods have revolutionized generative modeling, building upon early work in nonequilibrium thermodynamics (Sohl-Dickstein et al., 2015) and implicit models (Mohamed & Lakshminarayanan, 2017). Diffusion models were first proposed in DDPM (Ho et al., 2020), which shared a framework analogous to score-based models using Langevin dynamics Song & Ermon (2019). Subsequent innovations focused on improving sampling efficiency, with denoising diffusion implicit models (DDIMs) (Song et al., 2021) introducing a method to speed up inference with implicit modeling. Further advancements in accelerating the sampling process emerged through the application of stochastic differential equations (Song et al., 2020) and the development of numerical ODE solvers, exemplified by approaches like PNDM (Liu et al., 2021), significantly enhancing the practical utility of diffusion models in various generative tasks.

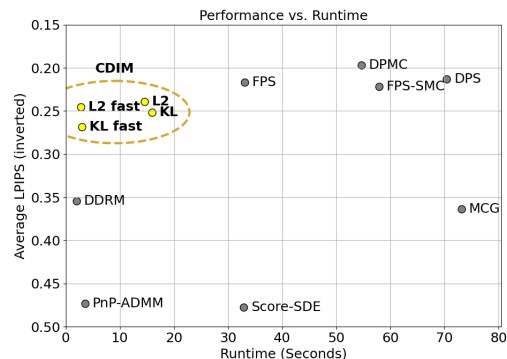

Figure 2: The inference speed and average LPIPS image quality score (inverted) averaged across multiple inverse tasks on the FFHQ dataset. The family of CDIM methods (top left corner) simultaneously achieves strong generation strong quality and fast inference, compared to other inverse solvers.

Applying diffusion models to inverse problems has been an active research area. DPS uses alternating projection steps to guide the diffusion process (Chung et al., 2022b). DDNM (Wang et al., 2022), DDRM (Kawar et al., 2022), SNIPS (Kawar et al., 2021), and PiGDM (Song et al., 2023a) use linear algebraic approaches and singular value decompositions. Techniques such as DMPS (Meng & Kabashima, 2022), FPS (Dou & Song, 2023), LGD (Song et al., 2023b), DPMC (Zhu et al., 2024), and MCG (Cardoso et al., 2023) focus on likelihood approximation for improved sampling. Guidance mechanisms were incorporated through classifier gradients (Dhariwal & Nichol, 2021), data consistency enforcement (Chung et al., 2022c), and low-frequency feature matching Choi et al. (2021). Other approaches use projection (Boys et al., 2023; Chung et al., 2024) or optimization (Chan et al., 2016; Wahlberg et al., 2012). DMPlug Wang et al. (2024) backpropagates through the entire diffusion process to optimize the noisy initialization $x_T$ so that the resulting output matches the observation. DSG (Yang et al., 2024) uses a similar

optimization update to us for enforcing consistency with the partial observation; however, it does not guarantee matching a constraint exactly, instead using a soft constraint, like DPS, to handle observational noise. Finally, works such as Blind DPS (Chung et al., 2022a) and FastEM (Laroche et al., 2023) solve inverse problems when the forward operator is unknown, a more difficult problem than the setting studied in this work.

## 3 BACKGROUND

We work in the context of DDPM (Ho et al., 2020), which models a data distribution $q(\mathbf{x}_0)$ by modeling a sequence $t = 1, \ldots, T$ of smoothed distributions defined by

$$q(\mathbf{x}_t|\mathbf{x}_0) = \mathcal{N}(\mathbf{x}_t; \sqrt{\alpha_t}\mathbf{x}_0, (1 - \alpha_t)\mathbf{I}). \tag{1}$$

The degree of smoothing is controlled by a monotone decreasing noise schedule $\alpha_t$ with $\alpha_0 = 1$ (no noise) and $\alpha_T = 0$ (pure Gaussian noise).[1] The idea is to model a *reverse process* $p_\theta(\mathbf{x}_{t-1}|\mathbf{x}_t)$ that that incrementally removes the noise in $\mathbf{x}_t$ such that $p_\theta(\mathbf{x}_T) = \mathcal{N}(\mathbf{x}_T; 0, \mathbf{I})$ and $p(\mathbf{x}_0)$ approximates the data distribution, where $p(\mathbf{x}_0)$ is the marginal distribution of outputs from the reverse process:

$$p_\theta(\mathbf{x}_0) = \int p_\theta(\mathbf{x}_T) \prod_{t=1}^{T} p_\theta(\mathbf{x}_{t-1}|\mathbf{x}_t) \, d\mathbf{x}_{1:T}. \tag{2}$$

Given noisy samples $\mathbf{x}_t = \sqrt{\alpha_t}\mathbf{x}_0 + \sqrt{1 - \alpha_t}\epsilon$, where $\mathbf{x}_0$ is a sample from the data distribution and $\epsilon \sim \mathcal{N}(0, \mathbf{I})$, a diffusion model $\epsilon_\theta(\mathbf{x}_t, t)$ is trained to predict $\epsilon$:

$$\min_\theta \mathbb{E}_{\mathbf{x}_t, \epsilon} \left[ \|\epsilon - \epsilon_\theta(\mathbf{x}_t, t)\|^2 \right]. \tag{3}$$

To parameterize the reverse process $p_\theta(\mathbf{x}_{t-1}|\mathbf{x}_t)$, DDIM (Song et al., 2021) exploits the Tweedie formula (Efron, 2011) for the posterior mean of a noisy observation:

$$\mathbb{E}[\mathbf{x}_0|\mathbf{x}_t] = \frac{1}{\sqrt{\alpha_t}} \left( \mathbf{x}_t - \sqrt{1 - \alpha_t}\nabla_{\mathbf{x}_t} \log q(\mathbf{x}_t) \right). \tag{4}$$

Using the denoising model $\epsilon(\mathbf{x}_t, t)$ as a plug-in estimate of the score function $\nabla_{\mathbf{x}_t} \log q(\mathbf{x}_t)$ (Vincent, 2011) we define the Tweedie estimate of the posterior mean:

$$\hat{\mathbf{x}}_0 \equiv \frac{1}{\sqrt{\alpha_t}} \left( \mathbf{x}_t - \sqrt{1 - \alpha_t}\epsilon_\theta(\mathbf{x}_t, t) \right) \approx \mathbb{E}[\mathbf{x}_0|\mathbf{x}_t]. \tag{5}$$

And we use this estimator to define a DDIM forward process $\mathbf{x}_{t-1} = f_\theta(\mathbf{x}_t)$ defined by

$$x_{t-1} = f_\theta(\mathbf{x}_t) = \sqrt{\alpha_{t-1}}\hat{\mathbf{x}}_0 + \sqrt{1 - \alpha_{t-1}} \left( \frac{\mathbf{x}_t - \sqrt{\alpha_t}\hat{\mathbf{x}}_0}{\sqrt{1 - \alpha_t}} \right). \tag{6}$$

Unlike DDPM, the forward process defined by Equation (6) is deterministic; the value $p_\theta(\mathbf{x}_0)$ is entirely determined by $\mathbf{x}_T \sim \mathcal{N}(0, \mathbf{I})$ thus making DDIM an implicit model.

With a slight modification of the DDIM update, we are able to take larger denoising steps and accelerate inference. Given $\delta \geq 1$, we define an accelerated denoising process

$$x_{t-\delta} = f_\theta^\delta(\mathbf{x}_t) = \sqrt{\alpha_{t-\delta}}\hat{\mathbf{x}}_0 + \sqrt{1 - \alpha_{t-\delta}} \left( \frac{\mathbf{x}_t - \sqrt{\alpha_t}\hat{\mathbf{x}}_0}{\sqrt{1 - \alpha_t}} \right). \tag{7}$$

Using this process, inference is completed in just $T' \equiv T/\delta$ steps, albeit with degraded quality of the resulting sample $\mathbf{x}_0$ as $\delta$ becomes large.

Diffusion Posterior Sampling (DPS) was an early work proposed applying diffusion models to solve inverse problems $\mathbf{y} = \mathbf{A}\mathbf{x}$ by alternating denoising steps with gradient descent on $\nabla_{\mathbf{x}_{t-1}} \|\mathbf{y} - \mathbf{A}\hat{\mathbf{x}}_0\|$ (Chung et al., 2022b). However, simply combining accelerated DDIM denoising steps with DPS-inspired gradient steps does not produce high quality outputs, instead resulting in blurry reconstructions (See Appendix B.3). Intuitively, the problem is that these gradient steps do not allow $\mathbf{A}\hat{\mathbf{x}}_0$ to converge quickly enough towards $\mathbf{y}$ under the accelerated denoising schedule of DDIM.

---

[1]We define $\alpha_t$ using the DDIM convention (Song et al., 2021); our $\alpha_t$ corresponds to $\bar{\alpha}_t$ in Ho et al. (2020).

## 4 METHODS

We are interested in solving linear inverse problems of the form $\mathbf{y} = \mathbf{A}\mathbf{x}$, where $\mathbf{y} \in \mathbb{R}^d$ is a linear measurement of $\mathbf{x} \in \mathbb{R}^n$ and $\mathbf{A} \in \mathbb{R}^{d \times n}$ describes the form of our measurements. For example, if $\mathbf{A} \in \{0,1\}^{n \times n}$ is a binary mask (which is the case for, e.g., in-painting or sparse recovery problems) then $\mathbf{y}$ describes a partial observation of $\mathbf{x}$. We seek an estimate $\hat{\mathbf{x}}$ that is consistent with our observations: in the noiseless case, $\mathbf{A}\hat{\mathbf{x}} = \mathbf{y}$. More generally, we seek to recover a robust estimate of $\hat{\mathbf{x}}$ when the observations $\mathbf{y}$ have been corrupted by noise. Given a noise distribution $r$, we seek to minimize $D_{\mathrm{KL}}(\hat{r} \parallel r)$, where $\hat{r}$ is the empirical distribution of $d$ residuals, e.g., $\mathbf{y} - \mathbf{A}\hat{\mathbf{x}} \in \mathbb{R}^d$, between noisy observations $\mathbf{y}$ and our estimates $\mathbf{A}\hat{\mathbf{x}}$.

We rely on a diffusion model $p_\theta(\mathbf{x})$ to identify an estimate $\hat{\mathbf{x}}$ that is both consistent with the observed measurements $\mathbf{y}$ and likely according to the model. In Section 4.1, we propose a modification of the DDIM inference procedure to efficiently optimize the Tweedie estimates of $\hat{\mathbf{x}}_0$ to satisfy $\mathbf{A}\hat{\mathbf{x}}_0 = \mathbf{y}$ during the diffusion process, resulting in a consistent and likely final result $\mathbf{x}_0$. In Section 4.2 we extend this optimization-based inference procedure to account for noise in the observations $\mathbf{y}$. In Section 4.3 we describe an early-stopping heuristic to avoid overfitting to noisy observations, which further reduces the cost of inference. Finally, in Section 4.4 we show how to set the step sizes for these optimization-based methods.

### 4.1 OPTIMIZING $\hat{\mathbf{x}}_0$ TO MATCH THE OBSERVATIONS

For linear measurements $\mathbf{A}$, the Tweedie formula for $\hat{\mathbf{x}}_0$ (and the corresponding plugin-estimate Equation (5)) extends to a formula for the expected observations:

$$\mathbb{E}[\mathbf{y}|\mathbf{x}_t] = \mathbf{A}\mathbb{E}[\mathbf{x}_0|\mathbf{x}_t] \approx \mathbf{A}\hat{\mathbf{x}}_0. \tag{8}$$

For noiseless observations $\mathbf{y}$, we propose a modification of the DDIM updates Equation (6) to find $\mathbf{x}_{t-1}$ that satisfies the constraint $\mathbf{A}\hat{\mathbf{x}}_0 = \mathbf{y}$. I.e., at each time step $t$, we force the Tweedie estimate of the posterior mean of $q(\mathbf{y}|\mathbf{x}_t)$ to match the observed measurements $\mathbf{y}$:

$$\underset{\mathbf{x}_{t-1}}{\arg\min} \quad \|\mathbf{x}_{t-1} - f_\theta(\mathbf{x}_t)\|^2$$
$$\text{subject to} \quad \mathbf{A}\hat{\mathbf{x}}_0 = \mathbf{y}. \tag{9}$$

We can interpret Equation (9) as a projection of the DDIM update $f_\theta(\mathbf{x}_t)$ onto the set of values $\mathbf{x}_{t-1}$ that satisfy the constraint $\mathbf{A}\hat{\mathbf{x}}_0 = \mathbf{y}$. The full inference procedure is analogous to projected gradient descent, whereby we alternately take a step $f_\theta(\mathbf{x}_t)$ determined by the diffusion model, and then project back onto the constraint $\mathbf{A}\hat{\mathbf{x}}_0 = \mathbf{y}$. We implement the projection step itself via gradient descent, initialized from $\mathbf{x}_{t-1}^{(0)} = f_\theta(\mathbf{x}_t)$ and computing

$$\mathbf{x}_{t-1}^{(k)} = \mathbf{x}_{t-1}^{(k-1)} + \eta\nabla_{\mathbf{x}_{t-1}}\|\mathbf{y} - \mathbf{A}\hat{\mathbf{x}}_0\|^2. \tag{10}$$

As $t$ approaches 0, $\hat{\mathbf{x}}_0$ converges to $\mathbf{x}_0$ and $\|\mathbf{y} - \mathbf{A}\hat{\mathbf{x}}_0\|^2$ becomes a simple convex quadratic, which can be minimized to arbitrary accuracy by taking sufficiently many gradient steps. This allows us to guarantee exact recovery of the observations $\mathbf{y} = \mathbf{A}\mathbf{x}_0$ in the recovered inverse $\mathbf{x}_0$.

For $t$ close to $T$, we face two conceptual challenges in optimizing Equation (9). First, for large $t$, no value $\mathbf{x}_t$ will satisfy $\mathbf{A}\hat{\mathbf{x}}_0 = \mathbf{y}$ and therefore the optimization is infeasible. Second, the estimate of the score function $\nabla_{\mathbf{x}_t} \log q(\mathbf{x}_t)$ using $\epsilon_\theta(\mathbf{x}_t, t)$ may be inaccurate; we risk overfitting to a bad plug-in estimate $\hat{\mathbf{x}}_0$. We illustrate both these claims by considering the Tweedie estimator Equation (5) in the case $t = T$. In this case, $\mathbf{x}_t \sim \mathcal{N}(0, I)$ is independent of $\mathbf{x}_0$ and therefore $\mathbb{E}[\mathbf{x}_0|\mathbf{x}_t] = \mathbb{E}[\mathbf{x}_0]$, the mean of the data distribution $q(\mathbf{x}_0)$. Unless $\mathbf{A}\mathbb{E}[\mathbf{x}_0] = \mathbf{y}$, the optimization is infeasible when $t = T$. Furthermore, we observe that when $t = T$, the plug-in estimator $\hat{\mathbf{x}}_0$ is not independent of $\mathbf{x}_t$ and $\hat{\mathbf{x}}_0 \neq \mathbb{E}[\mathbf{x}_0]$. This is indicative of error in the plug-in estimator, especially at high noise levels.

In light of these observations, we replace Equation (9) with a Lagrangian

$$\underset{\mathbf{x}_{t-1}}{\arg\min}\, \|\mathbf{x}_{t-1} - f_\theta(\mathbf{x}_t)\|^2 + \lambda\|\mathbf{y} - \mathbf{A}\hat{\mathbf{x}}_0\|^2. \tag{11}$$

We can interpret Equation (11) as a relaxation of Equation (9); the regularization by $\lambda\|\mathbf{y} - \mathbf{A}\hat{\mathbf{x}}_0\|^2$ is achieved implicitly by early stopping after $k = K$ steps of gradient descent. In contrast to projection, this Lagrangian objective is robust to both (1) the possible infeasibility of $\hat{\mathbf{y}}_0(\mathbf{x}_t) = \mathbf{y}$ and (2) overfitting the measurements based on an inaccurate Tweedie plug-in estimator.

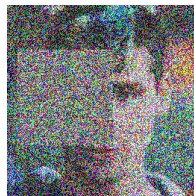 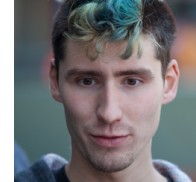 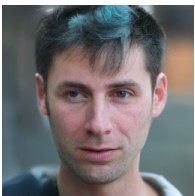 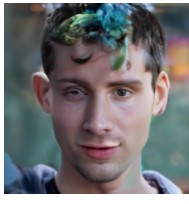 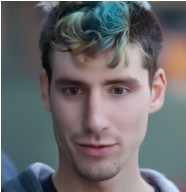

| Input: Box inpainting with bimodal noise | Ground Truth | DPS | Ours - L2 | Ours - Discrete KL |

Figure 3: Results on the box inpainting task with a bimodal noise distribution. By optimizing the discrete KL divergence, we can reconstruct the face with much higher fidelity than existing methods like DPS or our method with L2 loss.

### 4.2 OPTIMIZING THE KL DIVERGENCE OF RESIDUALS

For noisy inverse problems, imposing a hard constraint $\mathbf{A}\hat{\mathbf{x}}_0 = \mathbf{y}$ will overfit to the noise $\boldsymbol{\sigma}$ in the observations, as illustrated by Figure 4. Previous work accounts for noise using implicit regularization, by incompletely optimizing the objective $\mathbf{A}\hat{\mathbf{x}}_0 = \mathbf{y}$ (Chung et al., 2022b). In contrast, we propose to exactly optimize the Kullback-Leibler (KL) divergence between the empirical distribution of residuals $R(\mathbf{A}\hat{\mathbf{x}}_0, \mathbf{y})$ and a known, i.i.d. noise distribution $r$:

$$
\begin{aligned}
\arg\min_{\mathbf{x}} \quad & \|\mathbf{x} - \mathbf{x}_t\|^2 \\
\text{subject to} \quad & D_{\mathrm{KL}}(R(\mathbf{A}\hat{\mathbf{x}}_0, \mathbf{y}) \,\|\, r) = 0.
\end{aligned}
\tag{12}
$$

In Algorithm 1, we show how to optimize a constraint on categorical KL divergences to match arbitrary distributions of discretized residuals. We also provide a convenient objective for optimizing the empirical distribution of continuous residuals to match common noise patterns, including Gaussian and Poisson noise.

---

**Algorithm 1** Constrained Diffusion Implicit Models with KL Constraints

---

1: $\mathbf{x}_T \sim \mathcal{N}(\mathbf{0}, \mathbf{I})$
2: **for** $t = T, T - \delta, \ldots, 1$ **do**
3: $\quad \mathbf{x}_{t-\delta} \leftarrow \sqrt{\bar{\alpha}_{t-\delta}} \left( \frac{\mathbf{x}_t - \sqrt{1 - \bar{\alpha}_t} \epsilon_\theta(\mathbf{x}_t, t)}{\sqrt{\bar{\alpha}_t}} \right) + \sqrt{1 - \bar{\alpha}_{t-\delta}} \epsilon_\theta(\mathbf{x}_t, t)$  $\quad \triangleright$ Unconditional DDIM Step
4: $\quad$ **for** $k = 0, \ldots, K$ **do**
5: $\quad\quad \hat{\mathbf{x}}_0 \leftarrow \frac{1}{\sqrt{\bar{\alpha}_{t-\delta}}} \left( \mathbf{x}_{t-\delta} - \sqrt{1 - \bar{\alpha}_{t-\delta}} \cdot \epsilon_\theta(\mathbf{x}_{t-\delta}, t - \delta) \right)$
6: $\quad\quad \mathbf{x}_{t-\delta} \leftarrow \mathbf{x}_{t-\delta} + \eta \cdot \nabla_{\mathbf{x}_{t-\delta}} D_{\mathrm{KL}}(R(\mathbf{A}\hat{\mathbf{x}}_0, \mathbf{y}) \,\|\, r)$  $\quad \triangleright$ Projection
7: $\quad$ **end for**
8: **end for**
9: **return** $\hat{\mathbf{x}}_0$

---

**Additive Noise.** The general additive noise model is defined by $\mathbf{y} = \mathbf{A}\mathbf{x} + \boldsymbol{\sigma} \in \mathbb{R}^d$, where $\boldsymbol{\sigma} \sim r^{\otimes d}$. By discretizing the distribution of residuals into $B$ buckets, we can compute a categorical KL divergence between observed residuals and the discrete approximation of $r_B$ of $r$:

$$
D_{\mathrm{KL}}(R(\mathbf{A}\hat{\mathbf{x}}_0, \mathbf{y}) \,\|\, r_L) = \sum_{b=1}^{B} r_B(b) \log \left( \frac{r_B(b)}{\lfloor R(\mathbf{A}\hat{\mathbf{x}}_0, \mathbf{y}) \rfloor_B} \right).
\tag{13}
$$

In Figure 3 we show results on the box inpainting task when the observation has been corrupted with bimodal noise: $p(\boldsymbol{\sigma}_i = -0.75) = p(\boldsymbol{\sigma}_i = 0.75) = 0.5$ for $i = 1, \ldots, n$, where image pixels are normalized values $\mathbf{x}_i \in [-1, 1]$. We optimize the residuals using the discrete KL divergence and show that our result faithfully reconstructs the ground truth with high fidelity while filling in the missing section.

**Gaussian Noise.** Additive Gaussian noise is defined by $\boldsymbol{\sigma} \sim \mathcal{N}(0, \sigma^2 \mathbf{I})$, in which case the residuals $R(\mathbf{A}\mathbf{x}, \mathbf{y}) \equiv \mathbf{y} - \mathbf{A}\mathbf{x} \sim \mathcal{N}(0, \sigma^2 \mathbf{I})$ are i.i.d. with distribution $r \sim \mathcal{N}(0, \sigma^2)$. The empirical mean

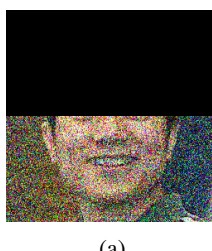 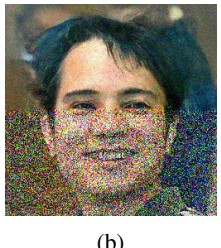 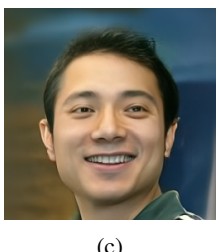

(a)  (b)  (c)

Figure 4: Results on a 50% noisy inpainting task. (a) is the noisy partial observation. (b) is generated by algorithm 2 without early stopping, showing that we can exactly match the observation even when the observation is out of distribution. (c) is generated by algorithm 2 with early stopping.

and variance of the residuals are

$$\hat{\mu} = \frac{1}{d} \sum_{i=1}^{d} R(\mathbf{A}\hat{\mathbf{x}}, \mathbf{y})_i, \quad \hat{\sigma}^2 = \frac{1}{d} \sum_{i=1}^{k} d \left( R(\mathbf{A}\hat{\mathbf{x}}, \mathbf{y})_i - \hat{\mu} \right)^2. \tag{14}$$

Using the analytical formula for KL divergence between two Gaussians ([Kingma & Welling, 2014](#)), we can match the empirical mean and variance of the residuals to $r$ by enforcing

$$D_{\mathrm{KL}}(R(\mathbf{A}\hat{\mathbf{x}}_0, \mathbf{y}) \parallel r) = \log\left(\frac{\sigma^2}{\hat{\sigma}^2}\right) + \frac{\hat{\sigma}^2 + \hat{\mu}^2}{2\sigma^2} - \frac{1}{2} = 0. \tag{15}$$

**Poisson Noise.** Possion noise is non-additive noise defined by $s\mathbf{y} \sim \mathrm{Poisson}(s\mathbf{A}\mathbf{x})$, where $\mathbf{y}$ is interpreted as discrete integer pixel values. The scaling factor $s \leq 1$ controls the degree of Poisson noise. Poisson noise is not identically distributed across $\mathbf{y}$; the variance increases with the scale of each observation. To remedy this, we consider the Pearson residuals ([Pregibon, 1981](#)):

$$R(\mathbf{A}\hat{\mathbf{x}}_0, \mathbf{y}) = \frac{\lambda(\mathbf{y} - \mathbf{A}\hat{\mathbf{x}}_0)}{\sqrt{\lambda\hat{\mathbf{x}}_0}}. \tag{16}$$

These residuals are identically distributed; moreover, they are approximately normal $r \sim \mathcal{N}(0, 1)$ ([Pierce & Schafer, 1986](#)). We can therefore optimize the KL divergence between Pearson residuals and a standard normal using Equation ([15](#)) to solve inverse problems with Poisson noise. Although the Pearson residuals closely follow the standard normal distribution for positive values of $\hat{\mathbf{x}}_0$, this breaks down for values of $\hat{\mathbf{x}}_0$ close to zero, and extreme noise levels $s$. In practice we find the Gaussian assumption to be valid for natural images corrupted by as much noise as $s \approx 0.025$. In Figure 1 we show an example of denoising an image corrupted by Poisson noise with $s = 0.05$.

### 4.3 NOISE-AGNOSTIC CONSTRAINTS

In many practical situations, we will not know the precise distribution of noise $r$ in the observations. For these cases, we propose a noise-agnostic version of CDIM, assuming only that the noise is zero-mean with variance $\mathrm{Var}(r)$. The idea is to directly minimize the squared error of the residuals, with early stopping to avoid overfitting to the noise once $\mathrm{Var}(r)$ exceeds the empirical variance of the residuals. In experiments, we find that this noise-agnostic algorithm performs similarly to the noise-aware versions described in Section 4.2. Moreover, the noise-agnostic algorithm is more efficient: by stopping early with enforcement of the constraint, it avoids excess evaluations of the model during the final steps of the diffusion process. The complete process is shown in Algorithm 2.

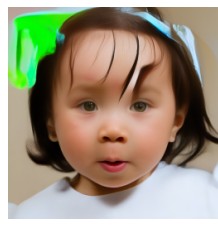 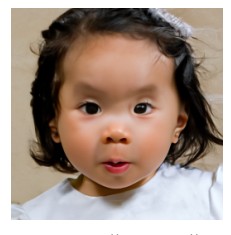 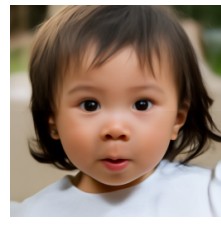

Input, 50% inpainting     $\eta \propto 1/\left\|\mathbf{y} - \mathbf{A}\hat{\mathbf{x}}_0\right\|$     $\eta \propto 1/\left\|\nabla_{\mathbf{x}_{t-\delta}}\right\|$     $\eta \propto 1/\mathbb{E}\left\|\nabla_{\mathbf{x}_{t-\delta}}\right\|$

Figure 5: Comparison of different step size schedules on a 50% inpainting task. We choose a challenging task with $T' = 10$, $K = 10$, $\sigma_y^2 = 0.15$ and use Algorithm 2. With enough steps, all three can produce reasonable results on $L^2$ optimization, but $\eta \propto 1/\mathbb{E}\left\|\nabla_{\mathbf{x}_{t-\delta}}\right\|$ is the most stable and converges the fastest.

---

**Algorithm 2** Constrained Diffusion Implicit Models with $L^2$ Constraints and Early Stopping

---

1: $\mathbf{x}_T \sim \mathcal{N}(\mathbf{0}, \mathbf{I})$
2: **for** $t = T, T - \delta.., 1$ **do**
3:      $\mathbf{x}_{t-\delta} \leftarrow \sqrt{\bar{\alpha}_{t-\delta}} \left( \frac{\mathbf{x}_t - \sqrt{1-\bar{\alpha}_t}\epsilon_\theta(\mathbf{x}_t,t)}{\sqrt{\bar{\alpha}_t}} \right) + \sqrt{1 - \bar{\alpha}_{t-\delta}}\epsilon_\theta(\mathbf{x}_t, t)$    $\triangleright$ Unconditional DDIM Step
4:      **for** $k = 0, .., K$ **do**
5:          $\hat{\mathbf{x}}_0 \leftarrow \frac{1}{\sqrt{\bar{\alpha}_{t-\delta}}} \left(\mathbf{x}_{t-\delta} - \sqrt{1 - \bar{\alpha}_{t-\delta}} \cdot \epsilon_\theta(\mathbf{x}_{t-\delta}, t-\delta)\right)$
6:          **if** $\hat{\sigma}^2 < \mathrm{Var}(r)$ **then**                $\triangleright$ Early Stopping
7:             **break**
8:          **end if**
9:          $\mathbf{x}_{t-\delta} \leftarrow \mathbf{x}_{t-\delta} + \eta \cdot \nabla_{\mathbf{x}_{t-\delta}} \frac{1}{d}\|R(\mathbf{A}\hat{\mathbf{x}}, \mathbf{y})\|_2^2$          $\triangleright$ Projection
10:      **end for**
11: **end for**

---

### 4.4 CHOICE OF STEP SIZE $\eta$

An important hyperparameter of these algorithms is the step size $\eta$. DPS sets $\eta$ proportional to $1/\|\mathbf{y} - \mathbf{A}\hat{\mathbf{x}}_0\|$ Chung et al. (2022b). We find that this fails to converge for KL optimization, and also produces unstable results for $L^2$ optimization when $T'$ is small. This is because $\|\mathbf{y} - \mathbf{A}\hat{\mathbf{x}}_0\| \to 0$ towards the end of the optimization, leading to extremely large steps. One option is to set $\eta$ inversely proportional to the magnitude of the gradient $\|\nabla\mathbf{x}_{t-\delta}\|$ at every single optimization step. Although this is the easiest solution, it can also result in unstable oscillations and slower convergence. Instead, we propose to set $\eta$ inversely proportional to $\mathbb{E}_{\mathbf{x}\sim\mathcal{X}_{\text{train}}}\left\|\nabla_{\mathbf{x}_{t-\delta}}\right\|$, a common optimization heuristic (Amari, 1998; Pascanu & Bengio, 2014). In Appendix A we describe how to compute this expectation. In Figure 5 we show qualitatively what happens with different $\eta$ schedules.

We find that for a specific optimization objective and task, the magnitude of the gradient $\|\nabla\mathbf{x}_{t-\delta}\|$ is highly similar across data points, datasets, and model architectures. While it is difficult to reason analytically about these magnitudes due to backpropagation through the network $\epsilon(\mathbf{x}_t, t)$, we empirically demonstrate this observation in Appendix A. This suggests that a learned step size based on $\mathbb{E}_{\mathbf{x}\sim\mathcal{X}_{\text{train}}}\left\|\nabla_{\mathbf{x}_{t-\delta}}\right\|$ generalizes as a good learning rate for unseen data. For all experiments, we estimate these magnitudes from FFHQ training data.

## 5 RESULTS AND EXPERIMENTS

We conduct experiments to understand the efficiency and quality of CDIM across various tasks and datasets. In Section 5.1, we present quantitative comparisons to state-of-the-art approaches, followed by ablation studies in Section 5.2 examining inference speed and hyperparameters. In Section 5.3 we explore two novel applications of diffusion models for inverse problems.

Table 1: Quantitative results (FID, LPIPS) of our model and existing models on various linear inverse problems on FFHQ $256 \times 256$-1k validation dataset. (Lower is better). The best result is in **bold** and the second best is underlined.

| FFHQ | Super Res | | Inpainting (box) | | Gaussian Deblur | | Inpainting (random) | | Runtime (seconds) |
|---|---|---|---|---|---|---|---|---|---|
| Methods | FID | LPIPS | FID | LPIPS | FID | LPIPS | FID | LPIPS | |
| Ours - KL fast | 36.76 | 0.283 | 35.15 | 0.2239 | 37.44 | 0.308 | 35.73 | 0.259 | 2.57 |
| Ours - $L^2$ fast | 33.87 | 0.276 | 27.51 | 0.1872 | 34.18 | 0.276 | 29.67 | 0.243 | 2.4 |
| Ours - KL | 34.71 | 0.269 | 30.88 | 0.1934 | 35.93 | 0.296 | 31.09 | 0.249 | 10.2 |
| Ours - $L^2$ | 31.54 | 0.269 | **26.09** | 0.196 | **29.68** | **0.252** | 28.52 | 0.240 | 9.0 |
| FPS-SMC | **26.62** | **0.210** | 26.51 | **0.150** | 29.97 | 0.253 | 33.10 | 0.275 | 116.90 |
| DPS | 39.35 | 0.214 | 33.12 | 0.168 | 44.05 | 0.257 | **21.19** | **0.212** | 70.42 |
| DDRM | 62.15 | 0.294 | 42.93 | 0.204 | 74.92 | 0.332 | 69.71 | 0.587 | 2.0 |
| MCG | 87.64 | 0.520 | 40.11 | 0.309 | 101.2 | 0.340 | 29.26 | 0.286 | 73.2 |
| PnP-ADMM | 66.52 | 0.353 | 151.9 | 0.406 | 90.42 | 0.441 | 123.6 | 0.692 | 3.595 |
| Score-SDE | 96.72 | 0.563 | 60.06 | 0.331 | 109.0 | 0.403 | 76.54 | 0.612 | 32.39 |
| ADMM-TV | 110.6 | 0.428 | 68.94 | 0.322 | 186.7 | 0.507 | 181.5 | 0.463 | - |

## 5.1 Numerical Results on FFHQ and ImageNet

We evaluate CDIM on the FFHQ-1k (Karras et al., 2019) and ImageNet-1k (Russakovsky et al., 2015) validation sets, both widely used benchmarks for assessing diffusion methods for inverse problems. Each dataset contains $256 \times 256$ RGB images scaled to the range [0, 1]. The tasks include 4x super-resolution, box inpainting, Gaussian deblur, and random inpainting. Details of each task are included in the appendix. For all tasks, we apply zero-centered Gaussian observational noise with $\sigma = 0.05$. To ensure fair comparisons, we use identical pre-trained diffusion models used in the baseline methods: for FFHQ we use the network from Chung et al. (2022b) and for ImageNet we use the network from Dhariwal & Nichol (2021). We use multiple metrics to measure the quality of the generated outputs: Frechet Inception Distance (FID) (Heusel et al., 2018) and Learned Perceptual Image Patch Similarity (LPIPS) (Zhang et al., 2018). All experiments are carried out on a single Nvidia A100 GPU.

In Table 1 we compare CDIM with several other inverse solvers using the FID and LPIPS metrics on the FFHQ dataset. We present results using both our KL divergence optimization method (Algorithm 1) and our $L^2$ optimization method (Algorithm 2) with early stopping. For these experiments, we present results with $T' = 50$ and $K = 3$ as well as $T' = 25$ and $K = 1$ labeled as "fast". For ImageNet results please see Appendix B.3.

## 5.2 Ablation Studies

**Number of Inference Steps.** CDIM offers the flexibility to trade off quality for faster inference time on demand. We investigate how generation quality changes as we vary the total computational budget during inference. Recall that the total number of network passes during inference is $T'(K + 1)$, where $T'$ is the number of denoising steps and $K$ is the number of optimization steps per denoising step. We use the random inpainting task on the FFHQ dataset with the setup described in the previous section. For this experiment we use KL optimization (Algorithm 1). The total network forward passes are varied from 200 to 20, and we show qualitative results. Notably, CDIM yields high quality samples with as few as 50 total inference steps, with quality degradations after that.

**T′ vs K trade-off**. We consider the optimal balance between $T'$ and $K$ when the total number of inference steps $T'(K + 1)$ is fixed. Using the random inpainting task on the FFHQ dataset with the previously described setup, we set $T'(K+1) = 200$ and analyze how PSNR, FID, and LPIPS change based on the chosen $T'$ and $K$ values. Results are plotted in Figure 7. FID results consistently favor the maximum number of denoising steps $T'$ with minimal optimization steps $K$. This is because FID evaluates overall distribution similarity rather than per-sample fidelity, and thus is not penalized by lower reconstruction-observation fidelity. In contrast, PSNR and LPIPS, which measure per-sample fidelity with respect to a reference image, achieve optimal results with a balanced mix of denoising and optimization steps.

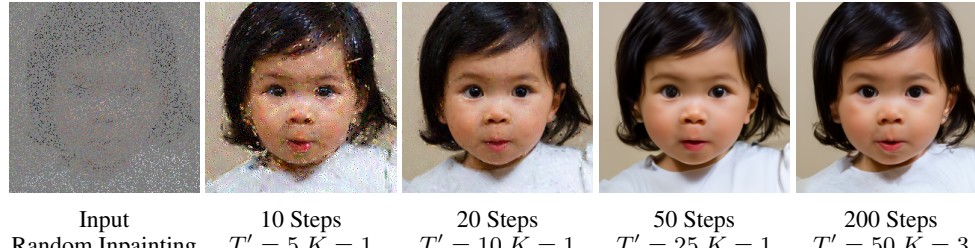

| Input Random Inpainting | 10 Steps $T' = 5\ K = 1$ | 20 Steps $T' = 10\ K = 1$ | 50 Steps $T' = 25\ K = 1$ | 200 Steps $T' = 50\ K = 3$ |

Figure 6: We reduce the total number of inference steps $T'(K + 1)$ and visualize the results. There is almost no visible degradation until less than 50 total steps.

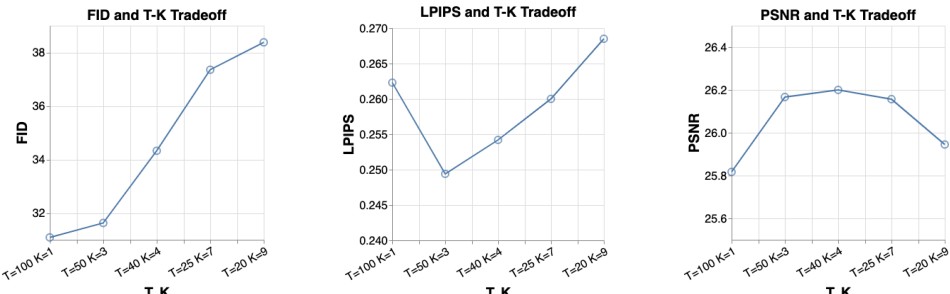

Figure 7: We fix the total number of inference steps at 200 and evaluate different combinations of T' and K. FID always prefers more denoising steps T', while LPIPS and PSNR are best at a mix of T' and K steps.

## 5.3 ADDITIONAL APPLICATIONS

**Time-Travel Rephotography** In Figure 1 we showcase an application of time-travel rephotography Luo et al. (2021). Antique cameras lack red light sensitivity, exaggerating wrinkles by filtering out skin subsurface scatter which occurs mostly in the red channel. To address this, we input the observed image into the blue color channel and use the pretrained FFHQ model with Algorithm 2 to project the face into the space of modern images. We further emphasize the power of our approach; Luo et al. (2021) trained a specialized model for this task while we are able to use a pretrained model without modification.

**Sparse Point Cloud Reprojection** For this task, 20 different images from a scene in The Grand Budapest Hotel scene were entered into Colmap (Schönberger & Frahm, 2016) to generate a sparse 3D point cloud. Note that the sparse nature of the Colmap point cloud means that projections of the point cloud will have roughly $90\%$ of the pixels missing. Furthermore, the observations often contain significant amounts of non-Gaussian noise due to false correspondences. We can formulate this as a noisy inpainting problem and use our method to fill in the missing pixels for a desired viewpoint. To address the errors in the point cloud, we use Algorithm 2 along with a variance threshold that adequately captures the imprecise nature of the point cloud. We showcase the results in Figure 8. Although this is not as robust as infilling the underlying point cloud directly, it does allow for realistic reprojections by infilling the sparse images.

## 6 CONCLUSION

In this paper we introduced CDIM, a new approach for solving noisy linear inverse problems with pretrained diffusion models. This is achieved by exploiting the structure of the DDIM inference procedure. By projecting the DDIM updates, such that Tweedie estimates of the denoised image $\hat{\mathbf{x}}_0$ match the linear constraints, we are able to enforce constraints without making out-of-distribution edits to the noised iterates $\mathbf{x}_t$. We note that our method cannot handle non-linear constraints, including latent diffusion, because for a non-linear function $h$, $\mathbb{E}\left[h(\mathbf{x}_0)\right] \neq h(\mathbb{E}\left[\mathbf{x}_0\right])$. Therefore, unlike the linear case of Equation (8), we cannot extend Tweedie's estimate of the posterior mean of $\mathbf{x}_0$ to an estimate of the posterior mean of non-linear observations $h(\mathbf{x}_0)$. However, for linear constraints,

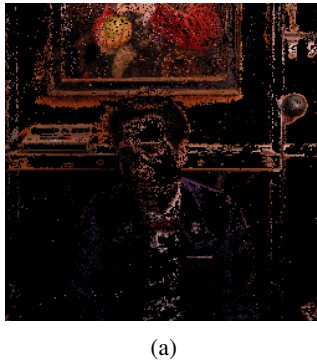
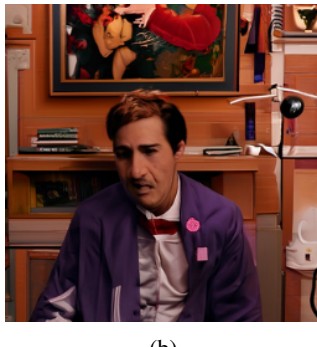

(a)                                    (b)

Figure 8: Using noisy inpainting to tackle sparse point cloud reconstruction. (a) Shows a sparse point cloud projected to a desired camera angle. (b) Shows the result after our method is used for noisy inpainting.

our method generates high quality images with faster inference than previous methods, creating a new point on the Pareto-frontier of quality vs. efficiency for linear inverse problems.

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

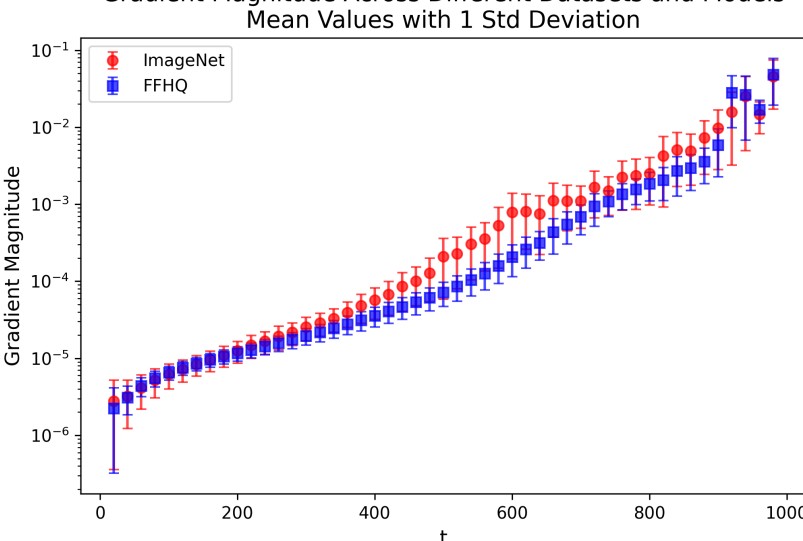

Figure 9: A plot of $\|\nabla_{\mathbf{x}_{t-\delta}}\|$ for two models and datasets, ImageNet and FFHQ. In each task 100 images were used. First, note the variance in a single task/model, shown by the error bars, is small. Second, note that the variance across the two tasks/models is also small.

# A  Calculating $\mathbb{E}\left\|\nabla_{\mathbf{x}_{t-\delta}}\right\|$

To calculate our expected gradient magnitude, we first start with simple gradient normalization: $\eta \leftarrow 1/\left\|\nabla_{\mathbf{x}_{t-\delta}}\right\|$, which normalizes our step size by the gradient magnitude on the fly at every optimization step. We run the full CDIM algorithm on the target task with the desired number of steps $T$ and $K$ on images from the training set. We calculate and store each gradient magnitude $\left\|\nabla_{\mathbf{x}_{t-\delta}}\right\|$ during the optimization process at every step. Finally, we average the empirical gradient magnitudes at each step $t - \delta$ to find $\mathbb{E}\left\|\nabla_{\mathbf{x}_{t-\delta}}\right\|$ across data points and inner optimization steps $k$. In practice we find that very few images are required to calculate a stable value for the expected gradient magnitude. In all experiments the value was calculated by running an initial optimization on 10 images from the training set.

# B  Additional Experimental Details

## B.1  Task Details

We describe additional details for each inverse task used in our experiments.

**Super Resolution** Images are downsampled to $64 \times 64$ using bicubic downsampling with a factor of 4.

**Box Inpainting** A random box of size $128 \times 128$ is chosen uniformly within the image. Those pixels are masked out affected all three of the RGB channels.

**Gaussian Deblur** A Gaussian Kernel of size $61 \times 61$ and intensity 3 is applied to the entire image.

**Random Inpainting** Each pixel is masked out with probability $92\%$ affecting all three of the RGB channels

**50% Inpainting** In various figures, we showcase a a $50\%$ inpainting task where the top half

of an image is masked out. This task is more challenging than box inpainting and can better illustrate differences between results.

## B.2 Measuring Runtime

To measure wall-clock runtime, we used a single A100 and ran all the inverse problems (super-resolution, box inpainting, gaussian deblur, random inpainting) on the FFHQ dataset. We only consider the runtime of the algorithm, without considering the python initialization time, model loading, or image io. For each task, we measured the runtime on 10 images and averaged the result to produce the final result. We note that the baseline runtimes are taken from Dou & Song (2023), where only the box inpainting task was considered. The runtime does not vary much between tasks when using CDIM, so we report our average runtime across tasks as a fair comparison metric.

## B.3 ImageNet Results

In Table 4 we report FID and LPIPS for the ImageNet dataset.

Table 2: Quantitative results (FID, LPIPS) of our model and existing models on various linear inverse problems on the Imagenet $256 \times 256$-1k validation dataset. (Lower is better)

| Imagenet | Super Resolution | | Inpainting (box) | | Gaussian Deblur | | Inpainting (random) | |
|---|---|---|---|---|---|---|---|---|
| Methods | FID | LPIPS | FID | LPIPS | FID | LPIPS | FID | LPIPS |
| CDIM - KL fast | 59.10 | 0.398 | 58.75 | 0.311 | 73.74 | 0.480 | 53.91 | 0.364 |
| CDIM - L2 fast | 53.70 | 0.378 | 52.00 | 0.267 | 56.10 | 0.393 | 51.96 | 0.370 |
| CDIM - KL | 47.77 | 0.347 | 48.26 | 0.2348 | 57.72 | 0.390 | 45.86 | 0.331 |
| CDIM - L2 | 47.45 | 0.339 | 50.31 | 0.251 | 38.69 | 0.347 | 46.20 | 0.332 |
| FPS-SMC | 47.30 | 0.316 | 33.24 | 0.212 | 54.21 | 0.403 | 42.77 | 0.328 |
| DPS | 50.66 | 0.337 | 38.82 | 0.262 | 62.72 | 0.444 | 35.87 | 0.303 |
| DDRM | 59.57 | 0.339 | 45.95 | 0.245 | 63.02 | 0.427 | 114.9 | 0.665 |
| MCG | 144.5 | 0.637 | 39.74 | 0.330 | 95.04 | 0.550 | 39.19 | 0.414 |
| PnP-ADMM | 97.27 | 0.433 | 78.24 | 0.367 | 100.6 | 0.519 | 114.7 | 0.677 |
| Score-SDE | 170.7 | 0.701 | 54.07 | 0.354 | 120.3 | 0.667 | 127.1 | 0.659 |
| ADMM-TV | 130.9 | 0.523 | 87.69 | 0.319 | 155.7 | 0.588 | 189.3 | 0.510 |

## B.4 PSNR Results

Table 3: Quantitative results (PSNR) of our model and existing models on various linear inverse problems on the FFHQ 256-1k validation dataset. (Higher is better)

| Imagenet | Super Resolution | Inpainting (box) | Gaussian Deblur | Inpainting (random) |
|---|---|---|---|---|
| Methods | PSNR | PSNR | PSNR | PSNR |
| CDIM - KL fast | 26.94 | 22.84 | 24.8 | 26.38 |
| CDIM - L2 fast | 27.08 | 23.20 | 26.77 | 26.49 |
| CDIM - KL | 27.11 | 23.54 | 25.68 | 26.97 |
| CDIM - L2 | 27.30 | 23.47 | 27.03 | 27.10 |
| FPS-SMC | 28.10 | 24.70 | 26.54 | 27.33 |
| DPS | 25.67 | 22.47 | 24.25 | 25.23 |
| DDRM | 25.36 | 22.24 | 23.36 | 9.19 |
| MCG | 20.05 | 19.97 | 6.72 | 21.57 |
| PnP-ADMM | 26.55 | 11.65 | 24.93 | 8.41 |
| Score-SDE | 17.62 | 18.51 | 7.21 | 13.52 |
| ADMM-TV | 23.86 | 17.81 | 22.37 | 22.03 |

Table 4: Quantitative results (PSNR) of our model and existing models on various linear inverse problems on the Imagenet 256 × 256-1k validation dataset. (Higher is better)

| Imagenet | Super Resolution | Inpainting (box) | Gaussian Deblur | Inpainting (random) |
|---|---|---|---|---|
| Methods | PSNR | PSNR | PSNR | PSNR |
| CDIM - KL fast | 23.17 | 19.64 | 21.26 | 21.95 |
| CDIM - L2 fast | 23.67 | 19.67 | 22.78 | 22.38 |
| CDIM - KL | 23.36 | 19.98 | 22.48 | 22.07 |
| CDIM - L2 | 23.92 | 20.06 | 23.32 | 22.61 |
| FPS-SMC | 24.78 | 22.03 | 23.81 | 24.12 |
| DPS | 23.87 | 18.90 | 21.97 | 22.20 |
| DDRM | 24.96 | 18.66 | 22.73 | 14.29 |
| MCG | 13.39 | 17.36 | 16.32 | 19.03 |
| PnP-ADMM | 23.75 | 12.70 | 21.81 | 8.39 |
| Score-SDE | 12.25 | 16.48 | 15.97 | 18.62 |
| ADMM-TV | 22.17 | 17.96 | 19.99 | 20.96 |

## B.5 COMPARISON WITH DPS USING DDIM

We show a qualitative comparison against DPS Chung et al. (2022b) when we use DDIM and fewer steps. We use the core DPS sampling algorithm, but with DDIM as the denoising algorithm instead of DDPM. The number of denoising steps is set to 50 and the step size of DPS is scaled to acheive the best convergence possible.

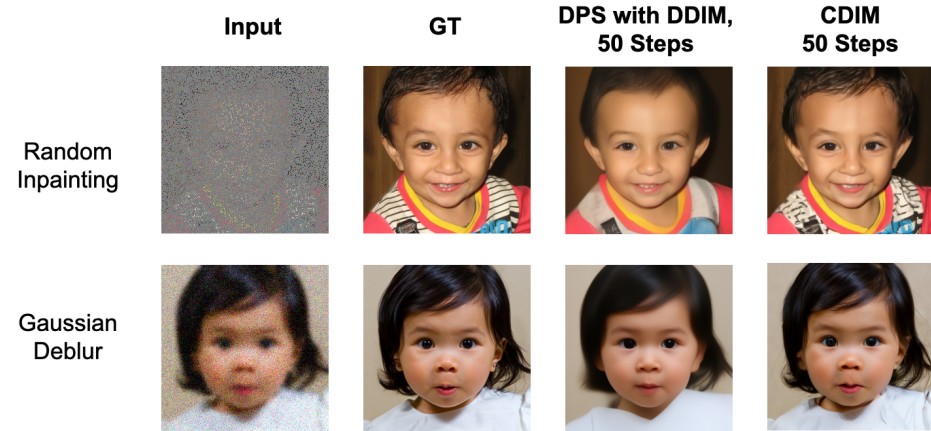

Figure 10: We show that our method is not simply DPS Chung et al. (2022b) with DDIM. If you just run DPS with DDIM and fewer steps, the output does not accurately match the observation. DPS ends up blurry and does not converge to match the constraint, and if you try to increase the step size it diverges. Our algorithm is able to accelerate inference better because we use a learned step size and use information about the underlying noise distribution.

## B.6 COMPARISON WITH DSG

We show a qualitative comparison against DSG Yang et al. (2024) on 3 tasks. We used the official code from their github, and generated results with 25 DDIM diffusion steps for both DSG and CDIM (and $K = 1$ for CDIM). As you can see, the DSG results are blurrier and sometimes contain artifacts

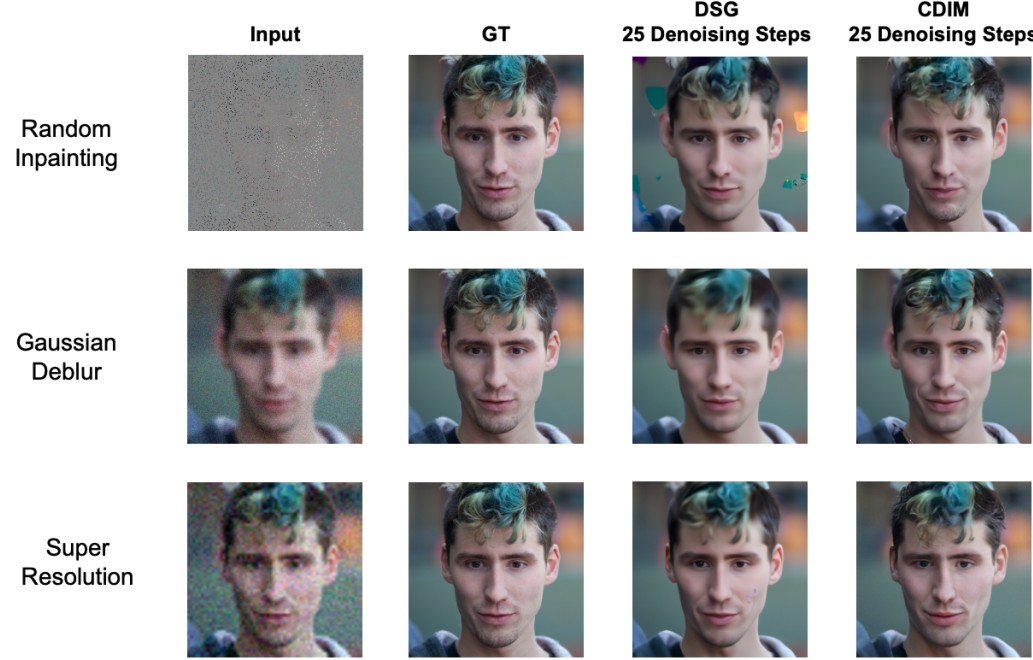

Figure 11: A comparison between DSG Yang et al. (2024) and CDIM when both algorithms use 25 DDIM denoising steps.

## B.7 EXTENDED RESULTS

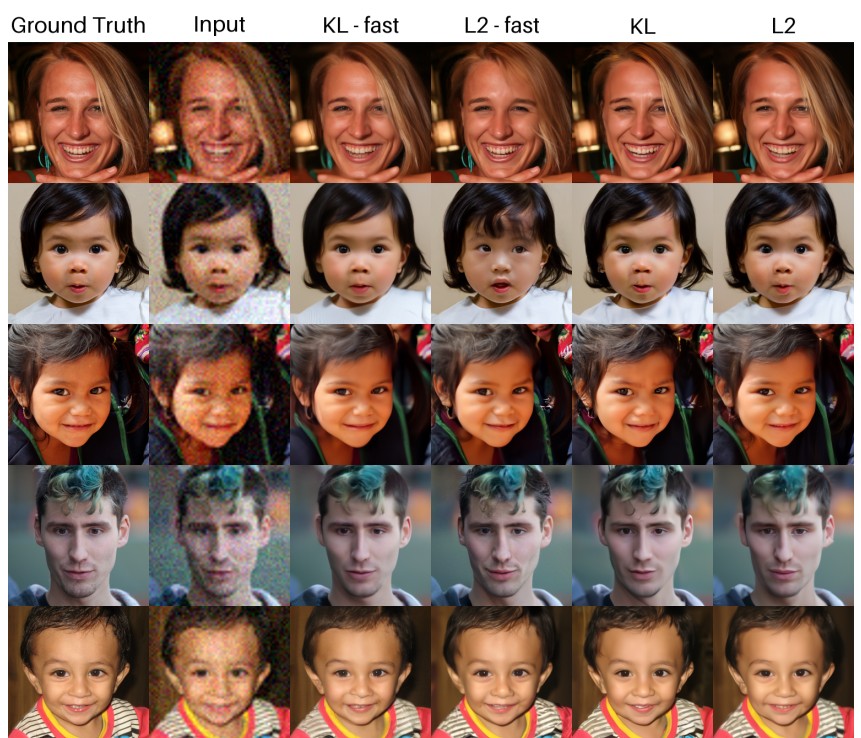

Figure 12: FFHQ Super-resolution extended results

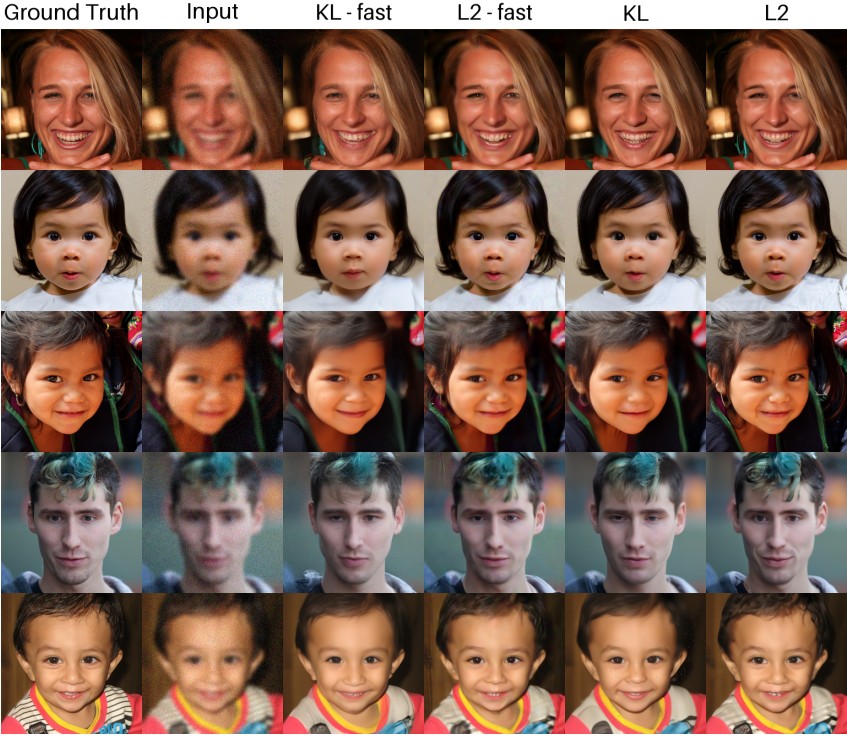

Figure 13: FFHQ Gaussian deblur extended results

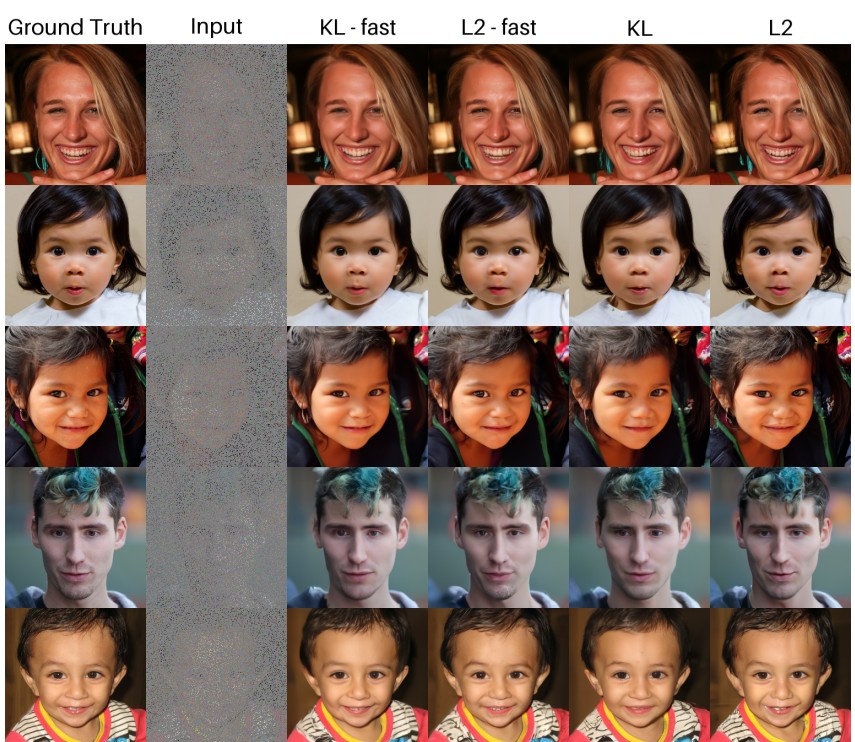

Figure 14: FFHQ random inpainting extended results

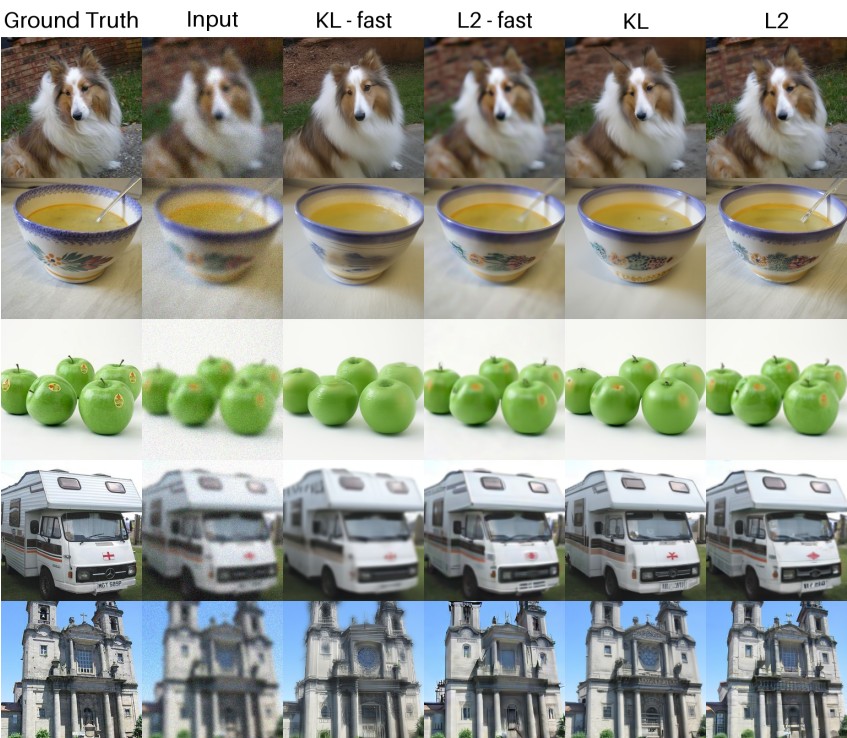

Figure 15: ImageNet Gaussian deblur extended results

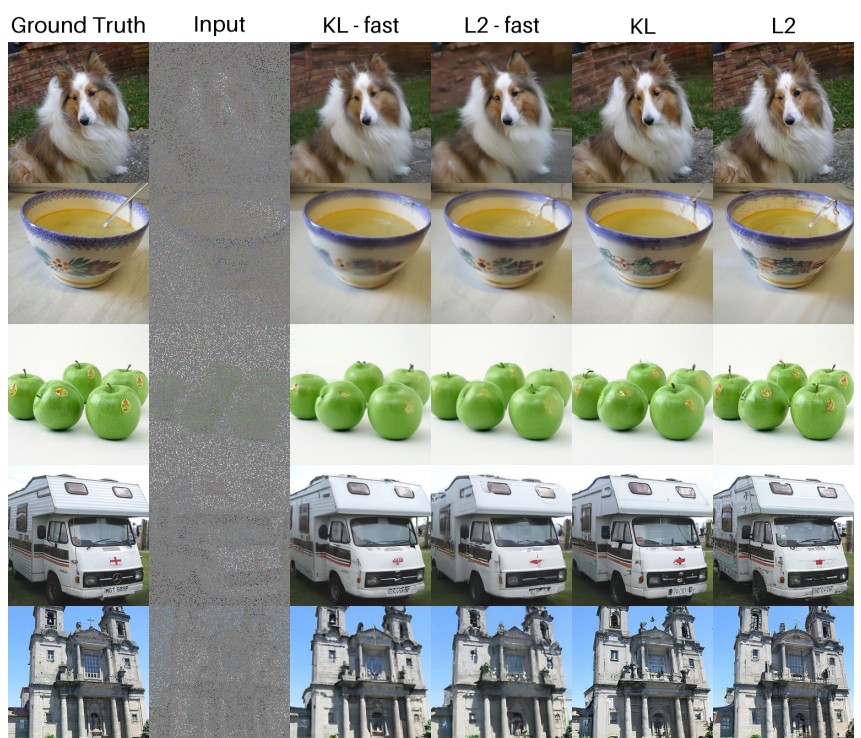

Figure 16: ImageNet random inpainting extended results

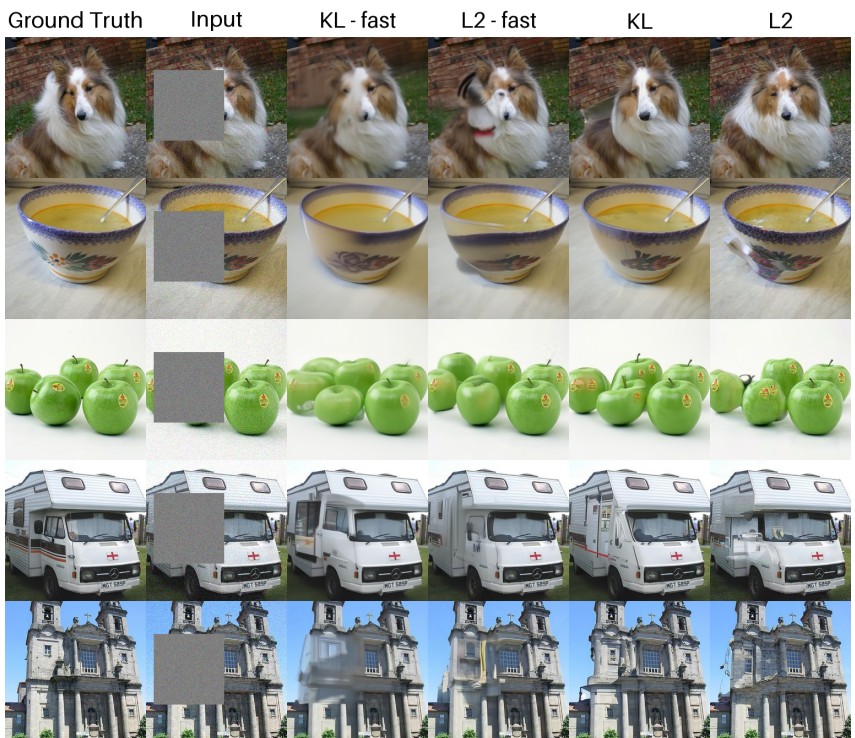

Figure 17: ImageNet box inpainting extended results

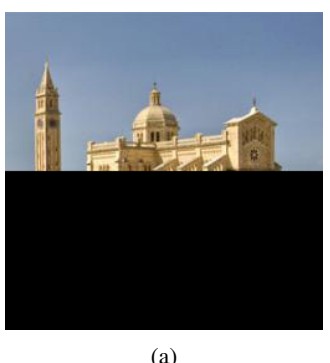 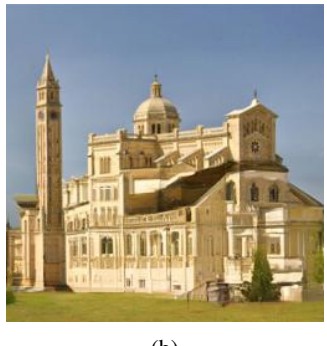

(a)                                                    (b)

Figure 18: Results on inpainting 50% of an image on LSUN Churches dataset.

