# OpenReview forum: "Constrained Diffusion Implicit Models"
_ICLR.cc/2025/Conference — Submitted to ICLR 2025_

### Official Review · Reviewer_cNaq · 2024-10-19

**Soundness:** 3
**Presentation:** 3
**Contribution:** 2
**Rating:** 5
**Confidence:** 4

**Summary:**

This paper proposes a linear non-blind inverse framework to solve inverse problems such as denoising, inpainting, and deblurring. The key contribution is the use of Denoising Diffusion Implicit Models (DDIM), which reformulates the diffusion process as a deterministic ODE, allowing it to bypass the full T sampling process. To ensure the denoised image aligns with the observed data, the method employs gradient projection to adjust the denoising trajectory. Additionally, a self-adaptive parameter control strategy is introduced to balance the data term and prior term dynamically. The approach significantly reduces inference time and demonstrates improved performance over Diffusion Posterior Sampling (DPS) across multiple applications.

**Strengths:**

+ Efficiency:

The framework leverages DDIM, which bypasses the need for computing all 1000 denoising steps. As a result, it achieves impressive inference speeds (e.g., 2 seconds vs. 70 seconds for DPS).

+ Improved Performance:

The method demonstrates better performance than DPS, as shown by FID scores, though it lacks evaluation through other metrics like PSNR.

**Weaknesses:**

I have several concerns regarding its baselines, claims, and equations.

+ Baselines:

DPS was a pioneering work that introduced diffusion priors for solving inverse problems. While this paper extends DPS by using DDIM model, the field has evolved rapidly. Recent advancements such as latentDPS (incorporating latent diffusion models), blindDPS (addressing blind inverse problems), and new methods like fastEM and other two arXiv works have shown improved performance using EM frameworks. The authors should include discussions about these recent developments and add comparisons with latentDPS or blindDPS, which have been available for over a year.

+ Claims and Contribution:

The paper's efficiency seems to primarily come from switching from DDPM to DDIM, which is a known method for speeding up inference by reducing the number of denoising steps. This makes the paper’s core contribution somewhat limited, as it largely inherits benefits from DDIM. I also question where the performance improvement over DPS originates. Does the improvement come solely from DDIM? Typically, acceleration comes with a trade-off in performance, so the authors should clarify the source of the performance gains over DPS.

+ 3D Claims:

The claim about "3D point cloud reconstruction" in the abstract is misleading. The paper focuses on 2D image completion, in the last figure, it is just projected points based 2D completion, which is far from true 3D reconstruction. The authors should rephrase this to more accurately reflect the work done. Additionally, the title could be clearer—something like "Solving Linear Inverse Problems with Constrained Diffusion Implicit Models" would better convey the focus of the paper.

+ Equations:

Some equations lack clarity. In Equations (6) and (7), it would be helpful to explicitly include $x_{t-1}$, for instance: $x_{t-1}=f_{\theta}(x_t)=...$. Additionally, the explanation between lines 195-202, which suggests that one cannot get $x_0$ from $x_t$, is confusing. In DPS, the use of Tweedie’s formula to estimate  $\hat{x_0}$ instead of $x_0$ is mentioned and widely adopted, and the authors should rewrite this section to provide a clearer explanation.

**Questions:**

See above.

---

> ### Author Response · Authors · 2024-11-18
> **Response to Review**
>
> Thank you for your thoughtful feedback. A top level comment has been created to address questions that multiple reviewers asked. A revision has been uploaded with changes in red. Below are our responses to your questions.
>
>
> Baselines - Thank you for sharing the other papers. We have added a discussion of Blind DPS and FastEM to the related work section in the revised paper. There are a large number of latent diffusion inverse methods, which are out of the scope of this paper.
>
> “The paper's efficiency seems to primarily come from switching from DDPM to DDIM” - Simply switching from DDPM to DDIM does not speed up *constrained* sampling in the same way. Even concurrent methods like DPMC that use DDIM have to use 200 steps to ensure the constraints are met. See the top level comment to all the reviewers for additional discussion of this point and a direct comparison to a naive DDIM implementation.
>
> “3D point cloud is misleading” - Thank you for pointing this out. We have modified the phrasing in the abstract, changed the title of that experimental section, also added a sentence about this limitation in the experiment section.
>
> Equations - Thank you for pointing this out. In the uploaded revision, we have fixed equations 6 and 7 with your suggestion.
>
> PSNR - We have included the PSNR tables in the appendix of the uploaded revised version

---

> > ### Comment · Reviewer_cNaq · 2024-11-25
> >
> > I thank the reviewer for the example provided in the general response, which partially solve my concerns.  However, as mentioned in the DPS + DDIM naive comparison study link, the proposed method requires additional information on noise distribution, therefore what happens if noise distribution is unknown and how could it be extended to other inverse problems? Moreover, I believe more explanations about why DPS+DDIM brings blurry results are needed to justify the motivation.

---

> > > ### Author Response · Authors · 2024-12-04
> > > **Response to Reviewer CNaq**
> > >
> > > In the 3D image reprojection example, we show an example where the noise distribution is both unknown and non-gaussian. We only assume that we have a bound on the variance. The result demonstrates the ability to handle unknown noise distributions.
> > >
> > > The DPS + DDIM results are blurry because DPS is a soft optimizer that does not enforce the constraint upon final output. Both the DPS step size and the objective result in finding a point close to the constraint, but not exactly satisfying it. In contrast, we run a hard optimization which converges faster even with observational noise.
> > >
> > > Are there other specific inverse problems of interest? We can handle any linear inverse problems, non-linear inverse problems suffer from an inaccurate Tweedie's estimate.

---

### Official Review · Reviewer_Pq9w · 2024-10-30

**Soundness:** 2
**Presentation:** 3
**Contribution:** 2
**Rating:** 5
**Confidence:** 4

**Summary:**

The paper presents conditional diffusion implicit models(CDIM), which modify the diffusion updates to enforce a constraint upon the final
output to solve noisy linear inverse problems. CDIM satisfies the constraints absolutely for noiseless inverse problems. For noisy case, the author use KL divergence distance to generalize CDIM to constrain the residual distribution of the noise. Compared to other solvers, the family of CDIM method achieve good quality and fast inference time for inverse problems on FFHQ and Imagenet 1k dataset.

**Strengths:**

1. propose a modification of the DDIM inference procedure to efficiently optimize the Tweedie estimates of $\hat{x_0}$ to satisfy $A\hat{x_0} = y$ during the diffusion process

2. propose to exactly optimize the Kullback-Leibler divergence between the empirical distribution of residuals $R(A\hat{x_0},y)$ and a known, i.i.d. noise distribution r to solve noisy inverse problems

3. give a new choice of $\eta$ to ensure the convergence for KL optimization and stable results of $L^2$ optimization

**Weaknesses:**

1. The results of DSG[1] on FFHQ and ImageNet datasets are not given. The DSG shows better reconstruction quality and faster inference time on FFHQ and ImageNet datasets.

2. The KL optimization method(Algorithm 1) is proposed to solve noisy linear inverse problems with known noise distribution. However, from Table.1 and Table.2, $L^2$ optimization has better performance than KL optimization in most tasks. The KL optmization seems meaningless.

3. The calculation of Var(r) are not shown clearly. The necessity of early stopping are not clarified. I doubt that early stopping cannot perform well in noise agnostic taks.

\[1].Yang, Lingxiao, et al. "Guidance with spherical gaussian constraint for conditional diffusion." arXiv preprint arXiv:2402.03201 (2024).

**Questions:**

1. Eq. (6), Eq. (7) have typo errors.
2. the Eq. (4) have deductive error, not $\sqrt{1-\alpha_{t}}\nabla_{x_t}\log q(x_t)$，should be ${(1-\alpha_{t})}\nabla_{x_t}\log q(x_t)$

---

> ### Author Response · Authors · 2024-11-18
> **Response to Review**
>
> Thank you for your thoughtful feedback. Below are our responses to your questions. A revision has been uploaded with changes in red.
>
> DSG - Thank you for pointing out DSG.  We have added a discussion of DSG to the related works section in the uploaded revision. Although their update step in the algorithm is similar, DSG does not guarantee matching a constraint exactly. Instead it uses a soft constraint, like DPS, to handle potential observational noise.
>
> We have also run a direct comparison against DSG when both algorithms use 25 DDIM denoising steps. [See the results here]( https://public-static-files.s3.us-west-1.amazonaws.com/DSG_comparison.png) and in the appendix B of the uploaded revision. You can see that DSG does not converge as well as CDIM with fewer steps, and the poor convergence of DSG with very few steps is also confirmed in page 14 of the DSG paper.
>
>
> KL seems meaningless - The L2 method outperforms KL in the table specifically because we are running experiments on gaussian noise where the L2 optimization well approximates the gaussian residual. We have modified Figure 3 to show L2 with early stopping for a highly non-gaussian noise example. It still produces a reasonable result, but optimizing the KL divergence is much better.
>
>
> Var(r) and early stopping - Var(r) is the variance of the observational noise distribution. Early stopping does indeed perform well in noise agnostic tasks and non-gaussian tasks. Figure 8 (3D point cloud reprojection) involves inpainting with an unknown noise distribution, and L2 with early stopping produces good results there.
>
>
> Thank you for pointing out typos in the equations which we have fixed

---

> ### Comment · Reviewer_Pq9w · 2024-11-23
>
> I doubt the definition of noise agnostic tasks. From my perspective, Var(r) is not known in noise agnostic tasks. Is there any citation to express the definition of noise agnostic tasks?

---

> > ### Comment · Reviewer_Pq9w · 2024-11-26
> >
> > I thank the author for the addtion content like comparison between CDIM and DSG. However, I doubt CDIM cannot handle unknown noise task very well.Hence, i maintain my initial score.

---

### Official Review · Reviewer_LUdH · 2024-11-01

**Soundness:** 2
**Presentation:** 2
**Contribution:** 2
**Rating:** 5
**Confidence:** 4

**Summary:**

The paper proposes a new approach for solving noisy linear inverse problems with pretrained diffusion models from the perspective of optimization. By leveraging the DDIM sampling process, it is more efficient than other diffusion based posterior sampling algorithms. It is capable of dealing with arbitrary noise in the observations.

**Strengths:**

1. The core contribution lies in combining the DDIM sampling process with an optimization perspective to maintain alignment between the posterior mean and the observation.
2. The paper is well-written and easy to follow, comprehensive experiments have been conducted.
3. The authors conduct thorough research on the efficiency and accuracy of different diffusion-based posterior sampling algorithms.

**Weaknesses:**

1. The contribution is limited, the idea of using DDIM to accelerate the sampling process is not new.
1. More mathematical deductions in the appendix would be helpful for the readers to understand. For example, Eq.13, and Eq.14. Also, an introduction to the diffusion posterior sampling(DPS) algorithm in the related work section is also helpful.
2. DMPlug[1] proposes a similar idea. The difference will be that their method optimizes the noise space. I would suggest a comparison with their method.

[1] Wang H, Zhang X, Li T, et al. DMPlug: A Plug-in Method for Solving Inverse Problems with Diffusion Models[J]. arXiv preprint arXiv:2405.16749, 2024.

**Questions:**

1. How much improvement does the optimization part make? Have the authors tried naive DDIM to solve inverse problems? What is the best result if we increase the number of optimization steps?

---

> ### Author Response · Authors · 2024-11-18
> **Response to Review**
>
> Thank you for your thoughtful feedback. Below are our responses to your questions. A revision has been uploaded with changes in red.
>
> “The idea of using DDIM to accelerate the sampling process is not new” - Please see our top-level comment to all reviewers, which addresses this point. While DDIM has been shown to accelerate unconstrained sampling, accelerating *constrained* sampling is not possible just by using DDIM naively. We have run new comparisons/experiments to demonstrate this.
>
> We have added a discussion of DPS into the background section to frame the problem better.
>
> DM Plug - We have added a discussion of DMPlug to our related works, thank you for pointing this method out. DM Plug takes around 10 minutes for inference, which is significantly slower than ours because they back-propagate through the entire diffusion process.
>
> Questions -
>
> Have the authors tried naive DDIM to solve inverse problems? Naive DDIM cannot solve inverse problems because the output will not satisfy the constraints. The optimization steps are necessary for the final output to satisfy the constraints, and our major contribution is showing how to satisfy the constraints while maintaining the acceleration of DDIM. In the top level comment we show the results of trying DPS with naively using DDIM.
>
> What is the best result if we increase the number of optimization steps? As you increase the number of optimization steps, the results improve up to a point at which it plateaus since the constraint is met. In general adding denoising steps is more effective to improving results than adding optimization steps.

---

> > ### Comment · Reviewer_LUdH · 2024-11-30
> >
> > I appreciate the authors for their feedback. I decide to maintain my evaluation as the contribution is limited to justify a higher score.

---

### Official Review · Reviewer_WmEo · 2024-11-03

**Soundness:** 2
**Presentation:** 2
**Contribution:** 2
**Rating:** 5
**Confidence:** 3

**Summary:**

This paper suggests a new model, Conditional Diffusion Implicit Models(CDIM) for solving linear inverse problem with pretrained diffusion models.
CDIM can address a problem whether it is noisy or not in linear cases. By imposing constraint on the prior diffusion objective, it solves a linear inverse problem efficiently in both time and utility.
Also for more efficient convergence, this paper utilizes Early Stopping and adaptive learning rate.
In experiments, it shows that it is fast, powerful and easy to use pretrained diffusion models without additional modules.

**Strengths:**

The authors have presented a promising approach, CDIM, that demonstrates notable improvements over existing methods. Key strengths include:

- **Efficiency**: The CDIM method shows a faster wall-clock time than other DDPM-based approaches (e.g., DPS) and achieves better performance metrics compared to DDIM-based methods (e.g., DDRM).
- **Exact Recovery for Noiseless Observations**: By incorporating the inverse relationship directly into the diffusion process as a constraint, the method can achieve exact recovery in the case of noiseless observations.
- **General Noise Model Applicability**: CDIM also addresses scenarios involving general noise models, broadening its potential use cases.

**Weaknesses:**

While the paper has several strengths, there are some areas where further clarification and refinement would enhance its impact and precision:

- **Early Stopping Criterion**: The paper suggests that the method handles unknown noise by utilizing early stopping based on the variance of residuals. However, the rationale for selecting the variance of residuals as an early-stopping criterion could benefit from a more detailed explanation. Additionally, the logical connection between noiseless methods (which aim to minimize KL divergence) and the noise-agnostic method (which minimizes squared error via early stopping) feels less cohesive. Further clarification in this section would strengthen the reasoning.
- **Accelerated Inference**: The paper mentions that CDIM achieves inference times 10 to 50 times faster than previous conditional diffusion methods. Could you clarify whether this acceleration is solely due to the use of DDIM, or if it represents a unique contribution of your own? Clearer differentiation here would improve understanding.

**Questions:**

1. **Naming Consistency**: The model is referred to as "conditional diffusion implicit models (CDIM)" within the text, yet the title uses "Constrained Diffusion Implicit Models." Additionally, there are existing conditional diffusion models, which may cause some confusion. Consistent naming throughout the paper could help to avoid this.
2. **Typos**:
    - Page 9, line 482: missing closing parentheses.
    - Page 13, line 672: "A. CALCULATIONG" (should be "CALCULATING").
    - Page 13, line 696: "A Gaussian Kernel of size '61x61' ~".
3. **Quantitative Results for Additional Applications**: For the Additional Applications section (Time-Travel Rephotography, Sparse Point Cloud Reconstruction), providing quantitative results would add further value and demonstrate the method's effectiveness.
4. **Highlighting Advantages of Using Only Pretrained Models**: The method reportedly improves certain aspects without additional modules, relying solely on pretrained models. Emphasizing this advantage more prominently could strengthen the appeal of the method.
5. **PSNR Measurements**: Table 1 currently lacks PSNR measurements. Including these would allow for more comprehensive performance assessment.
6. **Choice of Step Size in Section 4.4**: The paper notes that the DPS method fails in this context but doesn’t provide detailed reasons. A more thorough explanation here would be appreciated to clarify the underlying issues.

---

> ### Author Response · Authors · 2024-11-18
> **Response to Review**
>
> Thank you for your thoughtful feedback. Below are our responses to your questions. A revision has been uploaded with changes in red.
>
> Accelerated Inference - Please see the top level comment we created for all reviewers which addresses this question. DDIM by itself cannot accelerate inference to 50 steps for *constrained* sampling. Satisfying the constraints in accelerated sampling is a hard problem, and previous methods for constrained sampling fail to achieve the 10-50x speedups that DDIM achieves in the unconstrained setting (see Fig 1).
>
> Early Stopping Criterion - We will make this connection more clear. In the gaussian case, the KL divergence is minimized when the variance of the residuals is equal to sigma^2. For an unknown noise distribution, running L2 until the variance equals sigma^2 is the same as optimizing with a gaussian approximation. For noise that is poorly approximated by a Gaussian, e.g., multimodal noise distributions, L2 with early stopping produces poor results. We have updated figure 3 to show results of L2 with early stopping on a highly non-guassian noise distribution.
>
> Questions:
> Naming Consistency - We have uploaded a revised version where “constrained” diffusion implicit models is used everywhere
>
> Typos - Thank you for catching these. The typos are addressed in the uploaded revised version
>
> PSNR - We have included the PSNR tables in the appendix of the uploaded revised version
>
> Choice of Step Size - The step size in DPS is inversely proportional to ||Ax - y||. This quantity goes to 0, so the step size tends to get very large and unstable towards the end of the inference process. We borrow the idea of gradient normalization, which is a commonly used optimization technique and show empirically that it works better than the proposed step size in DPS. We have added an extra discussion about this in the revised paper.

---

### Author Response · Authors · 2024-11-18
**Response to All Reviewers**

We thank all the reviewers for their time and thoughtful feedback. We have attempted to answer all questions and run requested comparisons. We have uploaded a revision with changes highlighted in red.

Several reviewers have asked whether the speed up of CDIM comes from simply using DDIM instead of DDPM. Although DDIM greatly speeds up *unconstrained* sampling, simply using DDIM as a substitute for DDPM in *constrained* sampling does not speed up inference to a comparable level. Even concurrent works on inverse problems which use DDIM, such as DPMC [1] (presently under review at ICLR) still require 200 denoising steps for good results.

To further demonstrate this point, we have included another comparison study where we naively use DPS with DDIM updates, and show that it does not yield good results on its own. When we accelerate DPS using DDIM updates, the results are blurry and do not satisfy the constraints to an acceptable level. DPS does not even give results that exactly match constraints for non-accelerated DDPM updates. Furthermore, we show in Figure 3 that DPS fails for non-gaussian noise distributions.

[DPS + DDIM naive comparison study link](https://public-static-files.s3.us-west-1.amazonaws.com/DPS_Comparison.png)

This comparison is also included in the revised paper appendix B, and we have added a discussion of this to the background section.

We have responded to reviewer specific comments separately.


[1] Think Twice Before You Act: Improving Inverse Problem Solving With MCMC. Anonymous Authors. Submitted to The Thirteenth International Conference on Learning Representations

---

### Meta-Review · Area_Chair_WZBT · 2024-12-19

**Metareview:**

The main claim of the paper is that the proposed approach can solve linear inverse problems 10-50 times faster than existing works on conditional diffusion models. The achieved speed-ups are encouraging and were appreciated by the reviewers. However, the contribution over DDIM is incremental and the overall method is ad/hoc. Theoretical guarantees of constraint satisfaction (which are important in many linear inverse problems) could make this a stronger submission, and I encourage the authors to take the reviewers suggestions into account for a resubmission.

At this stage, I recommend to reject the work.

**Additional Comments On Reviewer Discussion:**

Reviewers found the contribution over DDIM to be incremental and limited, and this concern did not satisfactorily addressed in the rebuttal phase. On top of that, the paper seems to be lacking any theoretical guarantees which are important in inverse problems literature.

---

### Decision · Program_Chairs · 2025-01-22

Reject